# STAT3 Protein–Protein Interaction Analysis Finds P300 as a Regulator of STAT3 and Histone 3 Lysine 27 Acetylation in Pericytes

**DOI:** 10.3390/biomedicines12092102

**Published:** 2024-09-14

**Authors:** Gautam Kundu, Maryam Ghasemi, Seungbin Yim, Ayanna Rohil, Cuiyan Xin, Leo Ren, Shraddha Srivastava, Akinwande Akinfolarin, Subodh Kumar, Gyan P. Srivastava, Venkata S. Sabbisetti, Gopal Murugaiyan, Amrendra K. Ajay

**Affiliations:** 1Division of Renal Medicine, Department of Medicine, Brigham and Women’s Hospital, Boston, MA 02115, USA; 2US Military HIV Research Program (MHRP), Walter Reed Army Institute of Research, Silver Spring, MD 20910, USA; 3Henry M. Jackson Foundation for the Advancement of Military Medicine, Bethesda, MD 20817, USA; 4Department of Medicine, Harvard Medical School, Boston, MA 02115, USA; 5Institute of Science, Banaras Hindu University, Varanasi 221005, India; 6Department of Medical Oncology, Dana Farber Cancer Institute, Harvard Medical School, Boston, MA 02115, USA; 7Department of Electrical Engineering & Computer Science, University of Missouri, Columbia, MO 65211, USA; 8Ann Romney Center for Neurologic Diseases, Department of Neurology, Brigham and Women’s Hospital and Harvard Medical School, Boston, MA 02115, USA; 9Center for Polycystic Kidney Disease, Harvard Medical School, Boston, MA 02115, USA

**Keywords:** protein–protein interaction, histone acetylation, STRING, STAT3 acetylation, EP300, PIAS3, EGFR, pericytes

## Abstract

Background: Signal transducer and activator of transcription 3 (STAT3) is a member of the cytoplasmic inducible transcription factors and plays an important role in mediating signals from cytokines, chemokines, and growth factors. We and others have found that STAT3 directly regulates pro-fibrotic signaling in the kidney. The STAT3 protein–protein interaction plays an important role in activating its transcriptional activity. It is necessary to identify these interactions to investigate their function in kidney disease. Here, we investigated the protein–protein interaction among three species to find crucial interactions that can be targeted to alleviate kidney disease. Method: In this study, we examined common protein–protein interactions leading to the activation or downregulation of STAT3 among three different species: humans (*Homo sapiens*), mice (*Mus musculus*), and rabbits (*Oryctolagus cuniculus*). Further, we chose to investigate the P300 and STAT3 interaction and performed studies of the activation of STAT3 using IL-6 and inhibition of the P300 by its specific inhibitor A-485 in pericytes. Next, we performed immunoprecipitation to confirm whether A-485 inhibits the binding of P300 to STAT3. Results: Using the STRING application from ExPASy, we found that six proteins, including PIAS3, JAK1, JAK2, EGFR, SRC, and EP300, showed highly confident interactions with STAT3 in humans, mice, and rabbits. We also found that IL-6 treatment increased the acetylation of STAT3 and increased histone 3 lysine acetylation (H3K27ac). Furthermore, we found that the disruption of STAT3 and P300 interaction by the P300 inhibitor A-485 decreased STAT3 acetylation and H3K27ac. Finally, we confirmed that the P300 inhibitor A-485 inhibited the binding of STAT3 with P300, which inhibited its transcriptional activity by reducing the expression of *Ccnd1* (*Cyclin D1*). Conclusions: Targeting the P300 protein interaction with STAT3 may alleviate STAT3-mediated fibrotic signaling in humans and other species.

## 1. Introduction

Signal transducer and activator of transcription (STAT) proteins are members of the cytoplasmic inducible transcription factors that transfer extracellular signals from the cell surface to the nucleus, resulting in a target gene expression [1,2]. Among the seven STAT members that exist (STAT1–STAT4, STAT5a, STAT5b, and STAT6) [3], STAT3 was initially recognized through studies on acute response factor signaling [4]. Dysregulated STAT3 signaling has been implicated in various disease conditions, including cancers, autoimmune disorders, and kidney diseases [5,6,7]. The STAT3 protein consists of 770 amino acids and six domains with different functions, namely the N-terminal domain, coiled-coil, DNA binding, linker, Src homology-2 (SH2), and C-terminal transcription activation domain (TAD) [8,9]. The most well-known STAT3 signaling pathway is mainly activated by the binding of the IL-6 family cytokines to their receptors, which results in the homodimerization of gp130 or its heterodimerization with the other membrane receptors to transfer extracellular signals into the cell [10,11]. A common subunit of these receptor complexes is gp130. The dimerization of gp130 activates the associated Janus kinases (JAKs), and then JAKs phosphorylates tyrosine residues in the C-terminus of the gp130 [12]. JAK family kinases consist of four members, namely JAK1, JAK2, JAK3, and TYK2. Among these JAK family kinases, JAK1 and JAK2 are involved in the activation of STAT3 [13]. Alternative upstream activators of STAT3 signaling include non-receptor tyrosine kinases (e.g., Src and Bcr-Abl) and receptor tyrosine kinases (e.g., the EGF receptor, PDGF receptor, and VEGF receptor 2) [14,15].

The E1A-binding protein P300 (EP300, also known as p300) is a lysine acetyltransferase and a master regulator of gene transcription that promotes gene transcription in different cell types and is critical for various cellular functions such as proliferation, apoptosis, and differentiation [16]. P300 is a histone acetylase that binds to various proteins, leading to the acetylation of lysine residues and increasing protein activity. Reports suggest that STAT3 acetylation plays a critical role in and is required to form stable dimers. These STAT3 dimers, in response to cytokines such as IL-6, bind to transcription factors to cause fibrotic signaling [17]. P300 is known to acetylate the STAT3 on the Lysine 685 residue in kidney tubular epithelial cells, which is critical for developing a pro-fibrotic response [18].

The suppressor of the cytokine signaling (SOCS) protein family has a critical role in controlling STAT3 signaling. SOCS is induced after STAT activation by cytokines or growth factors and causes a negative feedback loop to inhibit the STAT activation [19]. SOCS1 downregulates STAT3 signaling by directly binding to JAKs and blocking the enzymatic activity of JAKs via an NH2-terminal domain sequence that resembles the JAK activation loop and acts as a pseudo substrate for the JAKs [20]. SOCS3, which contains a functionally equivalent NH2-terminal domain of SOCS1, blocks JAK activity by binding to cytoplasmic domains of receptors instead of the JAKsdirectly [21].

STAT3 is activated in various types of kidney diseases. According to a study conducted by Bertier et al., JAK1, 2, and 3 and STAT1 and STAT3 were expressed in patients with diabetic kidney disease (DKD) several-fold higher than those in normal controls [22]. In addition, cross-species transcriptional network analysis has indicated that JAK/STAT signaling is steadily upregulated in the glomeruli of patients with type 2 DKD, as well as three different murine models of DKD (streptozotocin DBA/2, C57BLKS *db*/*db*, and endothelial nitric oxide synthase-deficient C57BLKS *db*/*db* mice) [22,23]. A study conducted by Li et al. [24] demonstrated that the inhibition of STAT3 signaling via the natural flavonoid pectolinarigenin as a STAT3 inhibitor could reduce kidney fibrosis in a unilateral ureteral obstruction mice model by reducing the expression of multiple fibrotic genes, including *TGFB1* [25,26]. Furthermore, we found that STAT3 is activated in acute and chronic kidney diseases [27,28], and its deletion from pericytes protects mice from kidney fibrosis [29]. In addition, others have reported that JAK2-STAT3 is activated in polycystic kidney disease (PKD) and STAT3 inhibitors like pyrimethamine could inhibit ADPKD cell proliferation and alleviate disease progression in *Pkd1* mice [30,31,32].

Together, these preclinical studies suggest that JAK/STAT pathways play a major role in the pathogenesis and treatment of kidney diseases. However, there is no approved therapy to block JAK/STAT pathways in kidney diseases. In this study, we endeavored to identify a specific target of STAT3, common in three species (humans, mice, and rabbits).

For this, we analyzed STAT3 protein–protein interactions that are common in these three species. We found that P300 is a common interacting protein conserved in all three species. Thus, we investigated the role of P300 inhibition on STAT3 activation in pericytes and found that P300 directly binds to STAT3 for its acetylation.

## 2. Methods

### 2.1. STRING Analysis

**Confidence score distribution for each interaction source:** STRING database output provides confidence score measurement for each interaction source in addition to the combined score. We used the R/ggplot2 package to plot the score distribution. Based on these results, we chose interactions reported through only three sources, i.e., coexpression, pathways, and experimentally determined interactions.

**Organism-specific STAT3-binding proteins network analyses**: To ascertain the functional role of STAT3 in biological processes, an analysis of its interactome was carried out for three different organisms: humans, mice, and rabbits. We used the STRING database, consisting of all currently known proteins, comprising their structural attributes and predictions and their associations with other proteins based on functional characteristics such as gene coexpression, fusion, co-occurrence, location in genomic neighborhoods, and biological pathways, including experimentally determined results [33]. We used the STRING portal (https://string-db.org/) (access date 6 July 2024) for the network analysis of STAT3 biology. Using this portal, we selected three organisms, as mentioned, and used the full network, with the network type, experiment, database, and coexpression as active interaction sources; the minimum required interaction score was 0.7 (defined as high confidence). Graphical representations for the STAT3 network were obtained for the organisms, in which nodes represent individual proteins and edges signify interactions between them. The list of pathways in which STAT3 plays a significant role was obtained through the KEGG database [34,35]. The list of 25 top proteins was ranked based on the highest interaction strength.

**Cell culture and treatments:** 10T1/2 pericyte-like cells were purchased from ATCC (Cat. No. CCL-226, Manassas, VA, USA) and maintained in BME medium (Cat. No. 21010046, Thermo Fisher Scientific, Waltham, MA, USA) with 10% Fetal Bovine Serum (FBS) (Cat. No. A5256701, Thermo Fisher Scientific) and 2 mg/mL sodium pyruvate (Cat. No. 11360070, Thermo Fisher Scientific). Cells were grown in a 37 °C incubator with 5% CO_2_ and 37% relative humidity. IL-6 was purchased from R&D Systems (Cat. No. 206-IL-010, Minneapolis, MN, USA) and resuspended per the manufacturer’s instructions. A-485 was purchased from Tocris Biosciences (Cat. No. 206-IL-010, 6387, Bristol, UK).

### 2.2. Immunofluorescence Staining and Its Quantitation

Immunofluorescence staining: Immunofluorescence was performed as previously described [27,28,36]. Briefly, cells were treated with 10 ng/mL IL-6 with or without 10 μM of A-485 for 24 and 72 h. Vehicle-only treated cells were used as control. Cells were then fixed, with 4% PFA, for 15 min at room temperature. Samples were permeabilized using 1% Triton X-100 for 15 min. Two percent normal sheep serum containing 5% BSA was used for blocking for 1 h. Anti-AcSTAT3^K685^ (Cat. No. 2523, Cell Signaling Technology, Danvers, MA, USA) and anti-Histone H3 Lysine 27 (Cat. No. 8173, Cell Signaling Technology) were incubated overnight at 4 °C in five-times diluted blocking buffer containing 0.1% Tween 20. After three washes of PBST (0.1% Tween 20), anti-Rabbit Alexa 488-labeled secondary antibodies (Cat. No. A32790, Thermo Fisher Scientific) were added and incubated for 1 h at room temperature. Slides were washed and mounted with a medium-containing DAPI (Vector laboratories). Three to five images were captured using a 60X objective on a Nikon Confocal Imaging system (Nikon C1 Eclipse, Nikon, Melville, NY, USA). Experiments were conducted in triplicate and repeated two times.

The quantitation of immunofluorescence staining: Immunofluorescence staining was quantitated using Image J software (version 13.0.6). Cells were selected at a threshold of the same intensity for all the images, and masks were created. The integrated intensity and mean gray value were measured. Background intensity was calculated by selecting an area without cells. Corrected total cell fluorescence (CTCF) was calculated using the following formula: CTCF = integrated intensity − (area of selected cell × mean fluorescence of background readings). The CTCF values were normalized to the number of cells by counting DAPI-positive cells, and log10 values were plotted with ±SEM.

### 2.3. cDNA Extraction and RT-PCR

RT-PCR was performed as previously described [29]. Briefly, 1 × 10^5^ cells were plated in a 12-well plate and incubated overnight. Cells were then treated with 10 ng/mL of IL-6 with and without 10 μM A-485 for 24 h. RNA isolation was performed using TRIzol reagent (Cat. no. 15596026, Invitrogen Corporation, Waltham, MA, USA). Ten nanograms of RNA were used for making cDNA using a high-capacity reverse transcriptase kit (Cat. no. 4368814, Thermo Fisher Scientific). Two microliters of 1:5 diluted cDNA were used for RT-PCR. Gene-specific TaqMan probes for *Ccnd1* (Cat. no. Mm00232359_m1) and *Gapdh* (Cat. no. Mm99999915_g1) were used to amplify the specific transcripts. Fold change was calculated with respect to control cells and normalized to *Gapdh* using the delta Ct method.

### 2.4. Immunoprecipitation

Immunoprecipitation was performed as previously described [27,36,37]. Briefly, cells were treated with 10 ng/mL IL-6 with or without 10 μM of A-485 for 48 h and lysed in IP buffer (20 mM Tris.HCl, pH 8.0, 137 mM NaCl, 10% glycerol, 1% NP-40, and 2 mM EDTA) containing a protease inhibitor cocktail, and 300 μg protein was incubated overnight at 4°C with 10 μg of anti-STAT3 antibody (Cat. No. 12640, Cell Signaling Technology); 50 μL protein A/G agarose magnetic beads (Cat. No. 88802, Thermo Fisher Scientific) were added and incubated for 2 h at room temperature. The beads were washed three times with IP buffer in a magnetic rack. The immune complex was eluted by adding 1X SDS loading dye (Boston Bioproducts, Boston, MA, USA), followed by heating at 100 °C for 10 min, and immunoblotting was performed to detect STAT3 and P300.

### 2.5. Western Blotting

Western blotting was performed as previously described [27,29]. Immunoprecipitated samples were loaded onto the gel. Proteins were transferred onto the PVDF membrane and probed with specific antibodies. Anti-P300 (Cat. No. 57625) and anti-STAT3 (Cat. No. 9139) were purchased from Cell Signaling Technology. Anti-STAT3 antibody was used as a control for immunoprecipitation to confirm that the immunoprecipitation was successful. Blots were developed using Luminata Forte HRP reagent using a Syngene (Fredrick, MD, USA) imaging system.

### 2.6. Statistical Analyses

An analysis of variance was used to compare data among the groups. Student’s *t*-tests were used to determine the significant difference between the two groups. *p* values of less than or equal to 0.05 were considered significant. The results are presented as means ± SEM.

## 3. Results

### 3.1. STAT3 Functional Network Analysis Showed Higher Confidence for Text Mining, Database-Based, and Experimentally Verified Interactions

We performed STAT3 protein–protein interaction network analysis using the STRING database (http://string-db.org) (access date 6 July 2024), which provided a critical assessment and integration of protein–protein interactions, including direct (physical) as well as indirect (functional) associations. To explore all STAT3 interactors in humans genome-wide, unbiased to any in silico hypothesis or experimental validation, we extracted all interactor proteins from all evidence in the STRING database. We also extracted the confidence scores for each interaction based on evidence such as text mining, co-occurrence, a correlation based on RNA-seq data, etc. We found that text mining, database-based, and experimentally verified coexpression-based interactions show higher confidence scores (≥0.05) than co-localization-based evidence (Figure 1).

### 3.2. Analysis of Various Databases Shows Different Sets of Proteins Interacting with STAT3

We found 324 proteins with high confidence (total score) interacting with STAT3 (Appendix A). Therefore, to create the STAT3 network and extract the combined scores, we discounted text mining, homology-based, co-occurrence-based, and gene-fusion-based evidence. Based on this filtration, we narrowed the results down to 76 out of 324 proteins, which showed high-confidence interactions from multiple sources (Appendix A). We then categorized these STAT3-binding proteins based on their function by plotting them as a heatmap. We find four clusters: (a) non-homologous proteins interacting with STAT3 as experimentally determined and members of the same pathway; (b) STAT3-interacting proteins that are coexpressed with STAT3 and experimentally verified; (c) proteins that are experimentally determined, exist in similar pathways, and are homologous to STAT3; and (d) EP300 that is experimentally determined as a STAT3-interacting protein, coexpressed with STAT3 from RNA-sequencing data, and appears together in various pathways in databases (Figure 2).

### 3.3. Protein Interaction Analysis Using the STRING Database Finds P300 as a STAT3-Binding Protein Conserved among Three Species

Based on the current database, network graphs from STRING for the three selected organisms (human, mouse, and rabbit) show proteins having known or predicted interactions with STAT3. The data indicate 377 interactions for humans among 77 proteins, 192 interactions among 43 proteins for mice, and 163 interactions among 33 proteins for rabbits. They are, by default, selected according to a cut-off ‘score’ of = 0.7, which represents high-confidence interactions determined by the confidence levels of major sources of interactions, e.g., transcriptomics coexpression, pathway, and experimentally determined interactions found for the STAT3 protein in humans (Figure 3A), mice (Figure 3B) and rabbits (Figure 3C). The criteria for filtration and list of proteins in each node and cluster are shown in humans (Appendix A), mice (Appendix A), and rabbits (Appendix A). A list of proteins and their signaling pathways with their interaction strength and false discovery rate (FDR) for interactions with STAT3 have been analyzed from three independent databases: Reactome (Appendix A), KEGG (Appendix A), and WIKI (Appendix A). In these tables, the protein list is arranged from low to high FDR. Furthermore, we have included a complete list of STAT3-interacting proteins from these databases: KEGG (Appendix A), Reactome (Appendix A), and gene ontology biological process databases (Appendix A). The score for a given interacting protein ranges from 0 (lowest) to 1 (highest) and is indicative of the probability of links existing between it and STAT3. The results of proteins and interactions with scores above 0.7 are represented by the Venn diagram (Figure 3D). The Venn diagram shows 25 proteins common among all three organisms, that is, humans, mice, and rabbits. The list of these STAT3-interacting proteins is displayed in Figure 3E, which includes EP300.

### 3.4. P300 Inhibition Reduces STAT3 Acetylation and H3K27ac in Pericytes

To experimentally confirm the expression of STAT3 acetylation on the Lysine 685 residue and the acetylation of H3 lysine, we performed immunostaining in a time-dependent manner for 24 and 72 h. We utilized pericyte-like cells 10T1/2, which we and others have used to study STAT3 and other pro-fibrotic pathways in chronic kidney disease [29,38,39]. We treated these cells with inflammatory cytokine IL-6 to induce STAT3 activation. We found a significant increase in H3K27ac following IL-6 treatment, which was decreased by A-485 (Figure 4A). Furthermore, these cells showed a significant increase in STAT3 acetylation on the Lysine 685 residue following IL-6 treatment, which was inhibited by A-485 (Figure 4B). Further, we tested whether acetylation leads to the enhanced transcriptional activity of STAT3, and we performed RT-PCR to detect the *Ccnd1* (*Cyclin D1*) gene. Our results showed an increased expression of *Ccnd1* following IL-6 treatment that was decreased by A-485 treatment (Figure 4C). Thus, these data confirm that STAT3 acetylation increases the transcriptional activity of the *Cyclin D1* promoter. To confirm the binding of STAT3 with P300, we performed an immunoprecipitation assay using STAT3-specific antibodies. Our immunoprecipitation results showed a significant decrease in the binding of STAT3 and P300 after treatment with A-485 (Figure 4D). Immunoblotting for STAT3 served as a control for immunoprecipitation, showing that an equal amount of STAT3 was immunoprecipitated. Thus, these data confirm that IL-6 activation leads to an increase in the acetylation of both histone 3 Lysine 27 and STAT3 on the Lysine 685 residue. Finally, our data also confirm that P300 directly binds to P300 for its acetylation at Lysine 685.

## 4. Discussion

STAT3 is a family member of inducible transcription factors, and its dysregulation is pivotal in various kidney diseases in humans and their mouse models [22,27,30,40,41,42].

STAT3 signaling can be inhibited by a protein inhibitor of activated STAT3 (PIAS3). PIAS3 negatively regulates STAT3 and consequently targets gene expression by inhibiting the DNA-binding ability of STAT3 [43]. Considering the effect of PIAS3 on STAT3, the downregulation of PIAS3 expression may play a critical role in augmenting STAT3 signaling in kidney diseases and cancer development. In fact, studies have demonstrated that PIAS3 expression is reduced in various cancers, such as malignant melanoma, glioblastoma, lung squamous cell carcinoma, and anaplastic lymphoma [44,45,46,47,48]. On the other hand, the upregulation of PIAS3 expression in various tumors can inhibit cell proliferation and increase drug chemosensitivities such as lung cancer, prostate cancer, and glioblastoma [49,50,51,52,53].

The aberrant activation of EGFR leads to the deregulation of several downstream signaling cascades, including the STAT, MAPK, and AKT pathways [54]. There are four anti-EGFR medications approved by the FDA, which are cetuximab, panitumumab, gefitinib, and erlotinib, for treating certain cancers [55,56,57,58]. However, resistance to these medications is common by different mechanisms [59]. Some studies have suggested that the conjugation therapy of STAT3 inhibitors and EGFR blockade could result in encouraging outcomes [60,61,62,63]. For example, studies have shown that EGFR blockade alone exerts beneficial effects in progressive kidney disease, mainly ameliorating fibrosis [64,65]. However, the impact of combining EGFR-clocking drugs with STAT3 inhibitors in kidney diseases remains to be tested.

Src is a tyrosine kinase upstream of STAT3 and is frequently overactivated in different cancers [66,67,68]. Hamzeh et al. demonstrated that c-Src- and STAT3-dependent pathways cause cyclic stretch-induced TGF-β1 and fibronectin expression in renal epithelial cells. They also found that the inhibition of c-Src reduces STAT3 activation and fibronectin and TGF-β1 expression in HK-2 cells [69]. Pharmacological inhibition and the multi-kinase inhibitor tesevatinib result in the amelioration of disease phenotypes in various preclinical models of polycystic kidney disease (PKD) [70,71,72]. Xin et al. confirmed that sunitinib inhibits Src and STAT3, with no dramatic reduction in AKT, MAPK, and JAK signaling in 786-O and RCC4 tumor cells [73]. Wei et al. found that nobiletin (a poly-methoxy flavonoid isolated from *Citrus depressa* and *Citrus reticulata*) can decrease renal cell carcinoma volume and weight by reducing the levels of phosphorylated Src, phosphorylated AKT, serine/threonine kinase, and phosphorylated STAT3 [74]. Lue et al. found that combined Src-STAT3 inhibition using dasatinib and CYT387 (a JAK/STAT inhibitor) synergistically reduced cell proliferation and increased apoptosis in RCC cells [75]. Although STAT3 upstream kinase inhibitors such as AZD1480 showed preclinical efficacy, they failed in clinical trials due to a lack of clinical activity and toxicity [76]. STAT3 SH2 domain inhibitors, such as S3I-201, Stattic, and STA-21, which have successfully blocked STAT3 phosphorylation (Tyr705), dimerization, and DNA-binding activity, are either under development or unsuccessful [77,78,79]. Additional compounds have also been screened to inhibit the transcriptional activity of STAT3 to treat cancer and other diseases [80].

The E1A-binding protein P300 (EP300, also known as p300) is a lysine acetyltransferase and a master regulator of gene transcription that promotes gene transcription in different cell types and is critical for various cellular functions such as proliferation, apoptosis, and differentiation [16]. Chu et al. discovered that eGFR-associated CpGs have many binding sites for EBF1, EP300, and CEBPB. These results highlight regions in the locations in which DNA methylation plays a role in kidney function and CKD and suggest potential pathways important for kidney function in health and disease [81]. In a study conducted by Gong et al., it was indicated that *EP300* gene polymorphism correlates with the development and advancement of Diabetic Nephropathy (DN). In DN mice, silencing EP300 inhibited HIF2α expression levels and renal tubular fibrosis progression [82]. We and others have found that STAT3 activation leads to the progression of acute kidney injury and kidney fibrosis [27,29]. Reports confirm that the transformation of pericytes into myofibroblasts plays a crucial role in the development of chronic kidney disease [83,84]. Recently, we found that genetic depletion of STAT3 from pericytes protects mice from kidney fibrosis [29]. Targeting STAT3 protein–protein interaction can offer the specific targeting of STAT3 with a reduced off-target effect, as reviewed by Yeh et al. [85]. Our results reveal that P300 binding is critical for the acetylation of STAT3, which plays an important role in the development of pro-fibrotic signaling in pericytes. Based on the above, EP300 can be a promising candidate for preclinical testing in the field of kidney diseases and other indications. Thus, the present study identifies STAT3 interaction with P300 as a therapeutic target for developing fibrotic kidney disease involving the activation of pericytes. Our data show a significant decrease in H3K27ac at 72 h with the treatment of A-485 compared to 24 h. In addition, we found a significant reduction in STAT3 Lysine 685 acetylation at 72 h post A-485 treatment. Based on these data, we conclude that H3K27 and STAT3 Lysine 685 acetylation are not dependent on each other but rather independent. In addition, we found a decrease at the baseline at 72 h compared to 24 h of control for H3K27 acetylation. The significance of the P300-mediated acetylation of STAT3 leads to pro-fibrotic signaling and opens up a new therapeutic targeting strategy to alleviate fibrotic diseases. Based on the above findings, EP300 can be a promising candidate for future preclinical testing in the field of kidney diseases and other indications. Taken together, the present study identifies STAT3 interaction with P300 as a therapeutic target for the development of fibrotic kidney disease involving the activation of pericytes.

## Figures and Tables

**Figure 1 biomedicines-12-02102-f001:**
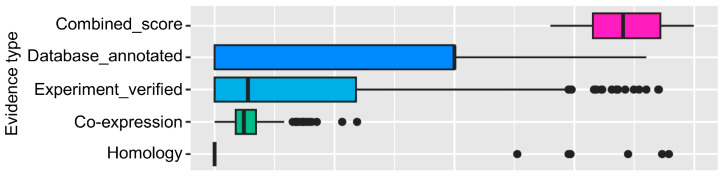
The boxplot shows interaction score distribution (*X*-axis) for STAT3-interacting proteins based on sources of evidence (*Y*-axis). We found that the aggregated score (combined_score) is mainly driven by text mining, which is known to have noise and false positive results. In addition, there is almost no evidence for homology. This suggests that major sources of identifying STAT3-interacting proteins are experimentally determined or appear in similar pathways. Some of these proteins also show coexpression statistically derived from RNA-seq data.

**Figure 2 biomedicines-12-02102-f002:**
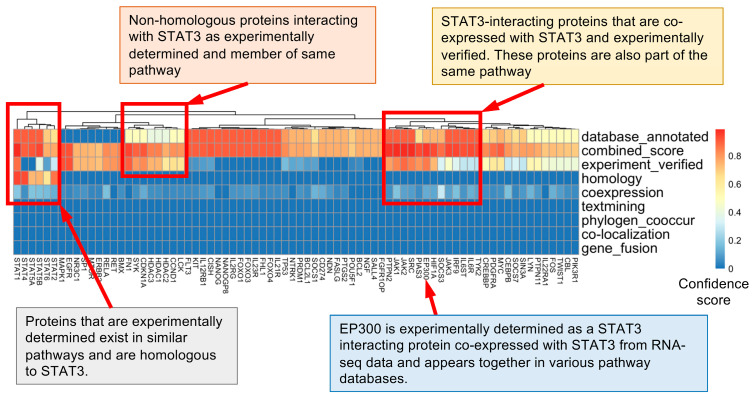
The heatmap shows the confidence score of every protein (shown as individual columns) estimated from each data source (shown as individual rows). The color represents the confidence score for each element in the metrics that ranges between [0, 1]. Clustering is performed on STAT3-associated proteins to group them based on scores from similar sources. Three groups of proteins are identified based on their association with STAT3 estimated from various sources. These sets of proteins are described in the heatmap.

**Figure 3 biomedicines-12-02102-f003:**
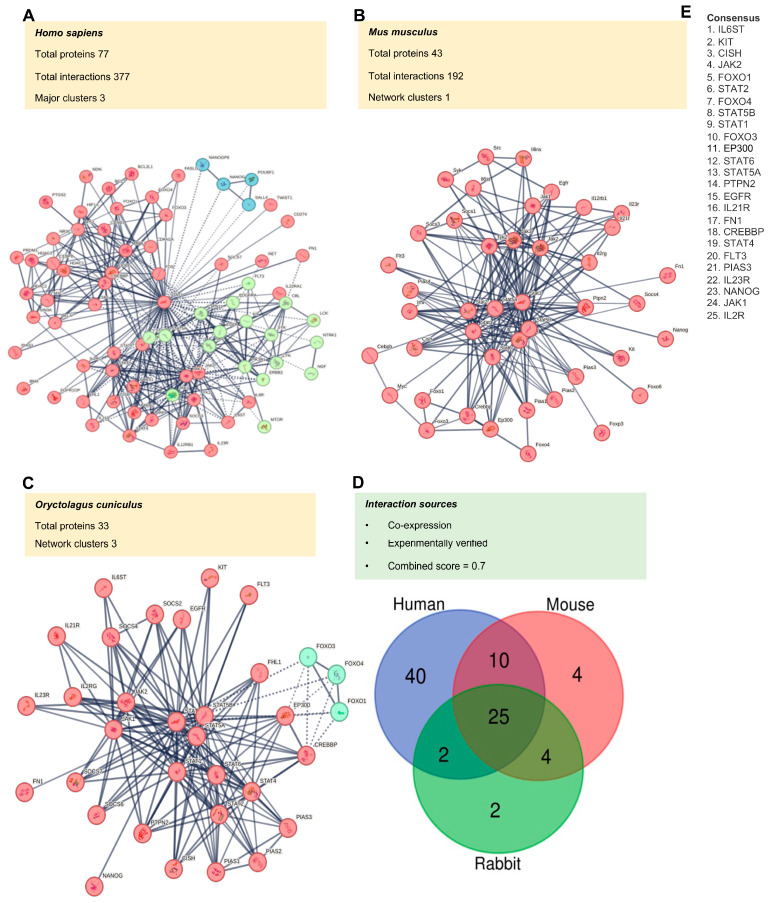
A protein–protein interaction network is created for STAT3-associated proteins from the STRING database. (**A**–**C**) Three different organisms (humans, mice, and rabbits) were selected to create the PPI network, where nodes are colored based on their cluster membership. Red represents the cluster with the largest number of proteins, green the cluster with a medium number, and blue the cluster with the smallest number of proteins. For each network, the annotation includes the total number of proteins interacting with STAT3, the total number of interactions amongst these proteins, and the total number of clusters identified. (**D**) The Venn diagram shows the overlap between these STAT3-interacting proteins in humans, mice, and rabbits. (**E**) The list of the top 25 proteins that are common in all three species is shown.

**Figure 4 biomedicines-12-02102-f004:**
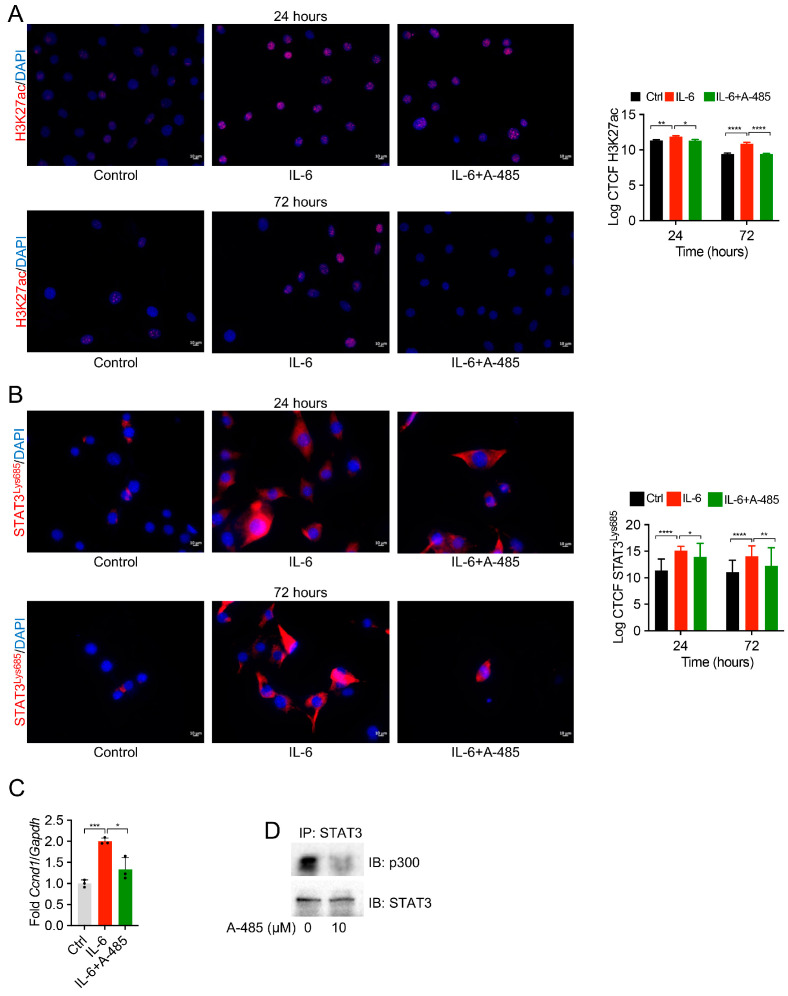
The inhibition of P300 inhibits IL-6-induced STAT3 acetylation and its binding. Immunofluorescence staining shows that (**A**) IL-6 treatment increased the Histone 3 Lysine acetylation (H3K27ac), which was decreased by the P300 inhibitor (A-485). The right panels show the quantitation of image intensity as log corrected total cell fluorescence (CTCF). Data are represented as ±SEM. (**B**) IL-6 treatment increased the acetylation of STAT3 on the Lysine 685 residue, which was inhibited by A-485 in pericytes, 10T1/2. Scale bar = 10 μm. The right panels show the quantitation of image intensity as log CTCF. The data are represented as ±SEM. (**C**) RT-PCR to detect *Ccnd1* (*Cyclin D1*) gene expression using RT-PCR. Fold changes relative to Ctrl were plotted after normalizing them to *Gapdh*. Data are represented as ±SD. (**D**) The immunoprecipitation of STAT3 and immunoblotting for P300 showed the binding of STAT3 with P300 and a decrease in P300 binding following A-485 treatment in pericytes, 10T1/2. * *p* ≤ 0.05, ** *p* ≤ 0.01, *** *p* ≤ 0.001 and **** *p* ≤ 0.0001.

## Data Availability

The original contributions presented in the study are included in the article/Appendix A, further inquiries can be directed to the corresponding author.

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
