# Peer review of "STAT3 Protein–Protein Interaction Analysis Finds P300 as a Regulator of STAT3 and Histone 3 Lysine 27 Acetylation in Pericytes"

_biomedicines, 2024, doi:10.3390/biomedicines12092102_

Round 1
Reviewer 1 Report (Previous Reviewer 1)
Comments and Suggestions for Authors
The authors present a manuscript where they show that an acetylation inhibitor A485 affects acetylation of STAT3 in pericytes that in turn impacts STAT3:p300 binding and potentially STAT3 transcriptional activity. The authors show a stringent way to identify interacting partners of STAT3 using publicly available data, however, given the nature of this approach, it hardly adds anything new to the field apart from it being done in pericytes. To be published, some stringency needs to be added to the experiment as mentioned below. My main concerns are time course and doses (points 5 and 6).
Comments to the authors:
1) Abstract needs to be re-written. The background section of the abstract should be directly relevant to the study and ideally finish with the knowledge gap that this study tries to fill.
2) Figure 1 is only marginally relevant to the study. I suggest putting it into supplementary material if the authors desire to keep it.
3) In figure 1, since the authors did not take into account “textmining”, “phylogeny_cooccur”, “co-localization”, “gene_fusion” these paramethers can be discarded from the figure to make it easier to understand.
4) In figure 1, the authors should also comment on other “clusters”/proteins with high confidence scores that they did not take into consideration. Is p300 cluster just one protein?
5) Considering the authors use rather long time points for IL6/A485 treatment (24 and 48 hours) while IL6 induces phosphorylation within minutes of treatment (at least in cancer cells), I think it would be beneficial to add a time course analysis which involves more time points (shorter than 24h) to see how STAT3 phosphorylation dynamics correlates with acetylation. This will help to determine whether the chosen time points for IP and stainings are relevant.
6) The authors should also add viability assays for the time points they use considering the high dose of the A485.
Comments on the Quality of English LanguageSome editing is required but it is not the most important thing at the moment
Author Response
Please see attached

Reviewer 2 Report (New Reviewer)
Comments and Suggestions for Authors
This manuscript describes an interaction between STAT3 and P300 by using bioinformatics and experimental validation. The authors found that P300 inhibitor A-485 could interrupt the binding of STAT3 and P300. The results are interesting. A few questions need to be answered.
1. Please confirm the number of network clusters in Figure 3B and 3C. It seems that figures are inconsistent with the information in related legends and Supplementary Figures. Also, would you please uniform the colors used in the 3 figures and provide a more detailed description of the colored nodes in the legends? It will make the figures more readable.
2. Please confirm the number of proteins in each cluster in Supplementary Figure 4 and 5, which is not consistent with the information provided.
3. What are the criteria for the ranking of the top 25 proteins in Figure 3D?
4. What does “CTCF” mean in the y-axis of Figure 4A and 4B? It will be more readable if three sub columns of 24 h and 72 h could be combined and compared respectively in the quantification. Is there any time-dependent manner of H3K27ac and STAT3 acetylation in Figure 4A and 4B? If so, please indicate it in the figures.
5. The result would be more convincing if a mutant of K685 could be used in the study. Moreover, did you test the percentage of STAT3 nuclear translocation during the immunofluorescence experiment? Also, the dimerization of STAT3 is also worth investigating with A-485 treatment.
Comments on the Quality of English LanguageThe legend of Figure in line 207-212 is described repeatedly in line 215-220. Also, please check the potential typo in the manuscript line 275 “48 h”.
Round 2
Reviewer 1 Report (Previous Reviewer 1)
Comments and Suggestions for Authors
Ok
Comments on the Quality of English LanguageOk
Author Response
None requested
Reviewer 2 Report (New Reviewer)
Comments and Suggestions for Authors
The author addressed the most questions in the response. While in the updated manuscript, the description in line 267-269 is contrary to the result shown in Figure 4. The author mentions a time-dependent effect of H3K27ac in line 263-264, please indicate the significance in the figure. Also, I’m still thinking that changing the data format will make the figure more easily accessible as I suggested in previous comment.
Comments on the Quality of English LanguageThere are several typos that need to be revised. Figure D should be E in Line 726. The P value format is also incorrect in Line 737 to 740.
Author Response
File attached

This manuscript is a resubmission of an earlier submission. The following is a list of the peer review reports and author responses from that submission.
Round 1
Reviewer 1 Report
Comments and Suggestions for Authors
Thanks you for inviting me to review this manuscript. The presented work bu Kundu et al. addresses a question of STAT3 interaction with an activating partner p300 and the implication of this interaction for renal diseases.
I am sorry to say that the paper needs some more work to be considered of value. I get the impression that the authors were in a hurry and hence the work is very raw. Please, see my comments below.
-
One of the strong points of this work is that the authors try to find a conserved interacting partner of STAT3 (among 4 species). However, they proceed to verify the interaction by the co-IP in a single cell line. To be consistent, the verification should be done in more species than just one.
-
The authors should show how IL6 induces STAT3 phosphorylation and acetylation and decreases methylation over time, not just in one static time point. This will allow to see whether p300 is necessary for the activation, or is it a secondary event. Consider doing WB so it is easy to follow a time course.
-
If immunostaining is used as main finding, the fluorescence intensity has to be quantified.
-
Most importantly: what are the functional consequences of p300 inhibition? Are any STAT3 target genes downregulated? This is an essential piece of data.
Minor points:
-
Language editing is advised throughout the manuscript.
-
Figures have to be better thought-through and have to be in a better resolution. Pay attention to the figure legends as well, they are not consistent.
-
Results should be better structured.
Extensive editing should be performed
Reviewer 2 Report
Comments and Suggestions for Authors
Gautam Kundu and colleagues present a quality and well-written experimental article focused on STAT3 protein-protein interaction analysis finds P300 as a regulator of STAT3 acetylation and Histone 3 lysine 27 trimethylation in pericytes.
Authors investigated the protein-protein interaction among four species to find crucial interactions that can be targeted to alleviate kidney disease.
Authors implicated common protein-protein interaction leading to activation or downregulation of STAT3 among four species Human (Homo sapiens), Mouse (Mus Musculus), Rabbit (Oryctolagus Cuniculus), and Rat (Rattus Norvegicus). Further, they chose to investigate the P300 and STAT3 interaction and performed activation of STAT3 using IL-6 and inhibition of the P300 by its specific inhibitor A-485 in pericytes. Next, they performed immunoprecipitation to confirm that the P300 inhibitor decreased the binding of P300 to STAT3. By STRING application from ExPASy, they indicated that six proteins, including PIAS3, JAK1, JAK2, EGFR, SRC, and EP300, showed highly confident interactions with STAT3 in humans, mice, rabbits, and rats.
Their results showed that IL-6 treatment increased acetylation of STAT3 and decreased H3K27 trimethylation. Further, disruption of STAT3 and P300 interaction by an inhibitor (A-485) of P300 decreased the STAT3 acetylation and restored H3K27 trimethylation. Further, they confirmed that the P300 inhibitor A-485 inhibited the binding of STAT3 with P300.
Finally, authors conclude that targeting the P300 protein interaction with STAT3 may alleviate STAT3-mediated fibrotic signaling in humans, mice, rabbits, and rats.
Overall, the manuscript is valuable for the scientific community and should be accepted for publication after edits are made.
===========================
Other comments:
1) Please check for typos throughout the manuscript.
2) Please improve figures and tables where appropriate.
3) With regards to PROTACs – authors are kindly encouraged to cite the following article that describes the novel approaches for the rational design of PROTAC linkers.
DOI: 10.37349/etat.2020.00023